# Dog Domestication Strongly Relied on Translation Regulation According to Differential Gene Expression Analysis

**DOI:** 10.3390/ani14182655

**Published:** 2024-09-12

**Authors:** David Jonas, Kitti Tatrai, Sara Sandor, Balazs Egyed, Eniko Kubinyi

**Affiliations:** 1Department of Ethology, ELTE Eötvös Loránd University, 1117 Budapest, Hungary; tatraikitti@gmail.com (K.T.); eniko.kubinyi@ttk.elte.hu (E.K.); 2MTA-ELTE Lendület “Momentum” Companion Animal Research Group, 1117 Budapest, Hungary; 3Department of Genetics, ELTE Eötvös Loránd University, 1117 Budapest, Hungary; egyed.balazs@ttk.elte.hu; 4ELTE NAP Canine Brain Research Group, 1117 Budapest, Hungary

**Keywords:** RNA sequencing, dog, wolf, domestication

## Abstract

**Simple Summary:**

This study investigates a genetic mechanism underlying the domestication of dogs from their common ancestors with wolves. RNA sequencing data were analyzed from the blood of both dogs and wolves, including samples from Europe Asia and North America, in order to identify differentially expressed genes (DEGs) and gene correlation networks. According to the results, 1567 DEGs, many of which were involved in translation regulation played a crucial role in dog domestication. The study highlights the importance of including diverse genetic samples in comparative analyses and enhances the understanding of the genetic basis of domestication.

**Abstract:**

Domestication of dogs from their shared ancestors with wolves occurred more than 15,000 years ago and affected many characteristics of the species. We analyzed the blood RNA sequence data of 12 dogs and 11 wolves from Europe and Asia to shed more light on the domestication history of dogs. We implemented a differential gene expression analysis, a weighted gene correlation network analysis, gene ontology and genetic pathway analyses. We found that both the sample origin (Europe or Asia) and the species had a significant effect on the blood gene expression profiles of the animals. We identified 1567 differentially expressed genes between wolves and dogs and found several significantly overrepresented gene ontology terms, such as RNA polymerase II transcription regulatory region sequence-specific DNA binding or translation. We identified 11 significant gene co-expression networks, hosting a total of 4402 genes, related to DNA replication, metabolism of RNA or metabolism of proteins, for example. Our findings suggest that gene expression regulation played a cardinal role in dog domestication. We recommend further diversifying the analyzed dog and wolf populations in the future by including individuals from different dog breeds and geographical origins, in order to enhance the specificity of detecting significant, true positive genes related to domestication as well as to reduce the false positive rate.

## 1. Introduction

The domestication of dogs (*Canis familiaris*) from their shared ancestor with grey wolves (*Canis lupus*) over 15,000 years ago marks one of the earliest and most significant human–animal relationships [1,2,3]. This process led to profound changes in the behavior, appearance, and physiology of dogs, distinguishing them from wolves and other wild canids. Despite extensive research into the genetic underpinnings of these changes, e.g., see [4,5,6], much remains to be understood about the specific genetic and transcriptomic shifts associated with domestication [7].

A key aspect of dog domestication involves understanding how gene expression differs between dogs and wolves. These differences likely contribute to the distinct traits observed in domesticated dogs, such as their enhanced sociability and altered stress responses. RNA sequencing (RNA-seq) analyzes differences between distinct groups of samples, such as breeds or species, which allows the quantitative analysis of gene transcription and the prediction of gene activity on a genome-wide scale [8]. Whole blood as a tissue sample is a good candidate for such an analysis as it has multiple advantages, e.g., see [9]. Sample collection is easy as standard procedures exist for sample collection, and importantly, it does not cause an unnecessarily high amount of pain or distress to the subject. Although the whole blood-derived RNA isolates could be hindered by the high abundance of hemoglobin-related messenger RNA, which can cause sensitivity-related issues in the downstream analyses [10], the hemoglobin reads can be removed in silico.

The first RNA-sequencing experiment from whole blood in dogs and wolves to investigate the protein-coding gene expression differences between the two species has been carried out by Yang et al. [11], who have provided insights into the transcriptomic differences between dogs and wolves. They identified several hundred differentially expressed genes and investigated their functions using the gene ontology (GO) and the Kyoto Encyclopedia of Genes and Genomes (KEGG) databases. Several of the significantly differentially expressed genes between dogs and wolves were related to the innate immune system based on their GO analysis. The authors also hypothesized that the higher aerobic capacity and hypoxia endurance ability in wolves—based on the published KEGG analysis—is related to the superior stamina of the wolves. However, the Yang et al. [11] study was based on a relatively small dataset containing two dog and three wolf samples.

Our primary aim is to investigate the whole-blood gene expression differences between dogs and wolves and shed more light on the domestication history of dogs. To obtain the most comprehensive and complete results possible, we implemented both a differential gene expression analysis and a weighted gene correlation network analysis on the RNA sequencing data as well as a gene ontology and pathway analysis to investigate the biological functions of the significant transcriptome-level differences. Our secondary aim was to compare our results to the previously published results, most notably to those of Yang et al. [11], because our current study is a continuation of theirs with a larger dataset and partly novel techniques. By including eighteen additional animals in the dataset (ten dogs from Jónás et al. [12] and eight wolves from Liu et al. [13]) we obtained a dataset that is more than four times larger than the previously analyzed one, adding significant statistical power. Consequently, we expect a more accurate and comprehensive prediction of the differentially expressed genes between the domesticated dog and its closest relative, the grey wolf.

## 2. Materials and Methods

### 2.1. Datasets

Below the four datasets used in this study are briefly introduced, with details presented only for the novel data. However, all details of the other three datasets are available in a comparative table as well as the similarities and differences between the bioinformatic analysis methods applied by Yang et al. and those applied here in Appendix A. The table provides all publicly available information and some additional details based on personal communication about sample collection, preservation, preparation and sequencing as well as about the bioinformatic analyses.


**Yang et al. (2018) [11]**


This dataset was downloaded from the [American] National Center for Biotechnology Information’s (NCBI) Sequence Read Archive (SRA) database (SRA IDs: SRP091691, SRP093404, SRP093423, SRP093543 and SRP093550; see Table 1 for sample ID correspondences). The dataset included three wolf and two dog samples from China. The wolf samples covered a large geographic region: two wolves were from Tibet (South China), while the third one was from Inner Mongolia (North China). The two dogs in this study were indigenous dogs from central China (Henan province). All individuals were unrelated, and their age is unknown. The dataset was described in more detail in Yang et al. [11].


**Jónás et al. [12]**


This dataset included ten unrelated pet border collie samples from our canine aging-related study (Jónás et al. [12]; data available on SRA under SRA BioProject ID: SRP367668; see Table 1 for sample ID correspondences). Blood samples were taken from ten unrelated border collies from two distinct age groups. The age of animals in the young group (*n* = 5) varied from 1 to 3 years (mean age: 1.4 years) and in the old group (*n* = 5) from 10 to 15 years (mean age: 12 years). The breed and the age of the dogs in our experiment were certified by their official documents and by their veterinarians. Based on the owners’ reports they were raised and kept as companion dogs in Hungary, they did not show any signs of illness and did not take any medications two weeks prior to the sampling. The animals’ health status was verified independently from the owners’ report by a veterinary diagnostic laboratory from 1 mL of blood sent immediately after sampling. Both the sex and the neuter status were mixed in both cohorts. Jónás et al. (present study and submitted) [12].

Three ml blood was taken from the vena cephalica or the vena saphena lateralis in accordance with the principles of lege artis by an experienced veterinarian in the presence of the dogs’ owners. It was collected directly into an RNA preservative fluid (DNA/RNABlood Collection Tubes, Zymo Research) and stored at −20 °C. Quick-DNA/RNA Blood Tube kits (Zymo Research, Irvine, CA, USA) were used to extract RNA. Possible DNA contamination was removed by DNase treatment before it was stored in a −80 °C ultra-low temperature freezer until further processing.

A poly(A) capture method was used to purify protein-coding RNA and TruSeq ^®^ Stranded mRNA Library preparation kit (Illumina, CA, USA) was used to perform library preparation. After DNase treatment and after library preparation a sample quality control was implemented. In all samples RNA integrity numbers varied from 8.5 to 10, indicating high-quality RNA materials.

Samples were sequenced at iBioScience (Pécs, Hungary) using a Novaseq 6000 Illumina sequencer machine. The minimum read number was set to 42 million paired-end reads per sample while the read length was 150 base pairs (bp). Sequence data came in fastq format from the company.


**Liu et al. (2017) [13]**


This dataset included eight wolf samples from China (Liu et al. [13]; data available on SRA under SRA IDs provided in Table 1). The dataset included four wolves from Inner Mongolia and four wolves from Tibet, all aged 3–4 years old.

Basic information about the samples and unique identifiers is presented in Table 1.


**Charruau et al. (2016) [14]**


Charruau et al. [14] analyzed whole blood RNA-seq samples from 27 wikves from Wyoming, the United States, out of which we selected 10 and included them in a multidimensional scaling analysis to investigate the effects of geographical origin on the gene expression profiles in whole blood transcriptome in more depth. This dataset was analyzed identically to the other three. The dataset of Charruau et al. [14] is available at NCBI’s Gene Expression Omnibus (ID: GSE80440).

### 2.2. Data Analysis

The ROS Cfam 1.0 reference genome of the domestic dog was used for the analysis with genome annotations from Ensembl v. 102 [15]. Gene biotype description sources were: Ensembl, GenCode [16] and Vega [17]. Apart from efficiency-enhancing parameters (e.g., increasing the number of processors or the amount of memory allowed to be used by a software), we aimed to use the default parameters of all software; if we had to adapt any default parameters to our dataset for any reason, it is explicitly mentioned and explained below.

### 2.3. Data Preparation and Alignment

Data analysis started with the quality control of the raw reads (FastQC software, version 0.12.0 [18]), followed by adapter sequence removal (cutadapt [19]). At this step, we also removed the trailing guanine (G) “bases” from the 3′ end of the reads, which are a result of the two-color sequencer technology (Novaseq sequencing system). The minimum read length was set to 50 bp at this step. Reads were aligned to the ROS Cfam 1.0 genome version of the domestic dog with HISAT2 using the dta option [20].

Following sequence alignment, all reads overlapping with hemoglobin genes were excluded from the analysis in silico, as recommended by Harrington et al. [10]. These represented only random noise in our analysis. We identified the canine homologs of the human hemoglobin genes published by Harrington et al. and removed all reads overlapping with those genes (see Appendix A for further information on these genes; the effects of the hemoglobin genes are further discussed in the Discussion section).

### 2.4. Gene Expression Analysis

A read count table was prepared from the aligned data with the featureCounts() function of the Rsubread R package, as recommended in the implemented DESeq2 workflow (see below). The following options were provided: isGTFAnnotationFile = T, isPairedEnd = T, countMultiMappingReads = F. Multi-mapping reads were excluded, following the requirements of the DESeq2 pipeline: “The matrix entries *K_ij_* indicate the number of sequencing reads that have been unambiguously mapped to a gene in a sample…” (quote from Love et al. [21]).

The DESeq2 R package was used to perform the differential gene expression analysis between the dog and wolf samples, following the workflow of Love et al., first published in 2015 [22]. According to this workflow, genes were kept if they were covered with at least ten reads in a minimum of eleven individuals (corresponding to the sample size of the smallest cluster in our analysis, the wolves).

According to the cited workflow, the regularized-logarithm transformation (rlog) possibly outperforms the variance stabilizing transformation (vst) in small samples (*n* < 30), and therefore, we used the rlog transformation to standardize the range of variances across samples for many descriptive statistical analyses of the read counts (this is also supported by an example plot of the transformed read counts on Appendix A).

Our model included, in addition to the intercept and species effects, a population effect or geographic origin, which was used to control for the significant effect of the sample origin (Europe vs. Asia). This effect was first observed in the descriptive statistical analysis of the raw data, where multiple different analyses (multidimensional scaling, principal component analysis, and sample distance calculations) all indicated a large effect of the sample origin, termed as population effect hereinafter. However, only the species effect was of interest to us and therefore, the population effect is not discussed beyond its direct effects on the analysis, i.e., its possible source, relevance, etc. is not investigated further. We also considered the possibility of including a breed effect in the model, but it was neither feasible nor separable from the above-mentioned population effect (these aspects are briefly discussed below in the Discussion section). We used the default parameter values when we identified the significantly differentially expressed genes, which were *p_adj_* < 0.1, where p_adj_ is the false discovery rate (FDR-) adjusted *p*-value.

A major advantage of the implemented analysis pipeline was that Yang et al. [11] also used the same R package (DESeq2) to analyze their data, and it allowed a direct comparison between the two studies (except for the differences in software versions).

We performed two independent analyses. Our primary aim was to analyze the dataset of Yang et al. [11] after adding our additional ten dog samples. Secondly, we also re-analyzed the data published by Yang et al. in 2018 (two dogs from Asia vs. three wolves from Asia), which we compared to the original results of Yang et al. as well as to our new results. This second analysis allowed us to negate some of the technical differences between the analysis of Yang et al. [11] and our analysis (e.g., software version differences).

Multidimensional scaling analysis using FPKM values for the European, Asian and American samples was performed as implemented in the PoiClaClu R package, using the PoissonDistance() and cmdscale() commands. Default parameter values were used for the former command, while the following were set for the latter: eig = TRUE and add = T.

After identifying the differentially expressed genes, we implemented a gene ontology analysis to investigate the functions affected by the identified genes [23,24], using the pantherDB online tool (version 19; [23,24]). For this analysis, PantherDB’s Slim database was used. In addition to the gene ontology analysis, we also made a weighted gene correlation network analysis (WGCNA) as implemented in the WGCNA R package [25] in order to cluster genes into modules based on their co-expression (or gene expression correlation) state. Modules are clusters of interconnected genes with high absolute correlation values (i.e., unsigned correlations). Eigengenes of each module “can be considered a representative of the gene expression profiles in a module” (quote and information from [25]). The input for this R package was prepared using the previously described DESeq2 package; DESeq2’s vst() function was used for normalization, followed by limma’s removeBatchEffect() function, which was used to remove the continent effect. Finally, the data matrix was transposed, as the WGCNA package requires an input matrix with samples in rows and genes in columns. Further parameter settings for WGCNA: the soft-thresholding power for network construction was set to 24 and a signed topological overlap matrix was used, according to the recommendations. Finally, a Reactome pathway analysis [26] was implemented on the significant gene co-expression sets, using the pantherDB v.19 on-line tool [27].

Tables and figures were created using basic R functions, frequently used R packages (ggplot2, pheatmap, VennDiagram, version 1.7.3.), and Microsoft Office.

## 3. Results

### 3.1. Descriptive Statistical Analysis

Table 2 shows basic alignment statistics of the studied 12 dog and 11 wolf samples. The sequencing depth was approximately 2.9 times higher in the ten European dogs (135 million reads on average) compared to the Asian samples (47 million reads on average). After adapter trimming and raw data filtering, the difference decreased (+116% more reads in the European samples). High sequence data quality was confirmed with the FastQC analysis, which was followed by read alignment to the reference genome. The alignment rate of the reads retained after adapter trimming was high, with an overall average of 92.3% (ranging from 87.4 to 96.4%). The difference between the Asian and European samples further decreased to approximately +100% after alignment to the ROS_Cfam 1.0 reference genome (Table 2).

A large proportion of the aligned reads mapped to the hemoglobin genes, ranging from 0.7 to 54.7% of the sequenced reads in the samples. The proportion of hemoglobin reads exceeded 40% in 4 out of the 23 samples. Interestingly, both Asian dogs were among those with the lowest number of hemoglobin-related reads in the samples.

Following the alignment to the reference genome, we identified 12014 expressed genes in our samples. Expressed genes were defined as having a minimum of ten reads per gene in at least eleven samples, in accordance with the Love et al. workflow [22]. This was performed to remove the genes with no information at all or with only a very limited amount of information in the studied samples.

A multidimensional scaling (MDS) analysis on the rlog-transformed raw read counts (Figure 1) was implemented, based on which both the population and species effects are evident, although the Asian dogs were located closer to the Asian wolves than to the European dogs. The results of a principal component analysis (PCA) as implemented in DESeq2 confirmed this pattern (Appendix A). We also calculated the Euclidean distances between the samples from the same transformed read counts, which analysis showed that the Asian dog samples clustered together with the Asian wolves (Appendix A). A similar distance matrix, but based on Poisson distances—which also takes into account the variance structure in the read counts—showed the Asian dogs to be more closely related to the European dogs than the Euclidean distances (Appendix A). Otherwise, the same differentiation could be observed in the sample distance clustering analysis as in the MDS and PCA analyses.

To further investigate the population effect, we downloaded ten wolf samples published by Charruau et al. [14] and included them in an MDS analysis together with the other analyzed samples. These wolves were from the United States, but their full inclusion in the study was not feasible due to major technical differences in the study design of Charruau et al. (most notably, their treatment of the samples with Globin Zero kit, removing globin-related RNA from the samples prior to sequencing) [14]. However, after the initial data procession (identical to that described above for our data) and the calculation of the number of fragments per kilobase transcript per million aligned fragments (FPKM) and extraction of the expressed genes, we could perform another multidimensional scaling analysis with additional wolf samples from the American continent. This analysis further supports that the blood RNA expression patterns are dependent on a strong population effect, i.e., the data points can be differentiated based on both the continent of origin (Asia, Europe, America) and the species (wolf, dog; Appendix A). On this figure, Asian dogs were closer to the European dogs than to the Asian wolves and one of the wolves seem to be an outlier.

### 3.2. Differential Gene Expression Analysis

We identified 1576 differentially expressed genes (DEGs) between the dogs and wolves after correcting for the population effect in the model. Out of these genes, 550 were downregulated in wolves compared to dogs, and 1026 were upregulated. There were 367 downregulated genes with at least 50% gene expression reduction in wolves and 807 with at least a 2-fold increase in gene expression in wolves (Figure 2 and Appendix A).

As expected, due to the implemented poly-A capture step in the experimental laboratory RNA processing, most of the DEGs coded proteins (*n* = 1502; 95% of all DEGs), including five immunoglobulin-related genes that undergo somatic recombination, as defined by the Ensembl biotypes. We also identified a small number of pseudogenes (*n* = 10), long non-coding RNAs (*n* = 47), and a small number of other types of non-coding RNA (*n* = 13; Appendix A).

The protein-coding genes with the lowest fold changes (<10% expression level; FDR adjusted *p*-values: 6.9 × 10^−7^–2.5 × 10^−4^) were: ENSCAFG00845013426 (CD5 molecule-like), ENSCAFG00845003711 (aggrecan), ENSCAFG00845022316 (radical S-adenosyl methionine domain containing 2), ENSCAFG00845006392 (kinesin family member 18B) and two novel genes (ENSCAFG00845009655 and ENSCAFG00845010005).

On the other hand, the most upregulated, known protein-coding genes in the wolves compared to dogs were ENSCAFG00845008386 (tubulin gamma 2), ENSCAFG00845011929 (receptor accessory protein 6) and ENSCAFG00845030690 (TMA7), with *log*_2_FC range: 6.5–8.7; range of FDR adjusted *p*-values: 6.0 × 10^−16^–8.2 × 10^−3^. Some novel genes (ENSCAFG00845020234, ENSCAFG00845005471, ENSCAFG00845008711) had estimated *log*_2_ fold change values higher than 10; however, their functions are unknown.

Most of the genes with low *p*-values had high normalized read counts, which increases the confidence in our differential gene expression analysis results (Appendix A).

### 3.3. Comparison with the Published Results of Yang et al. [11]

We re-analyzed the dataset of Yang et al. (*n* = 2 dog and *n* = 3 wolf samples) to obtain results that are directly comparable to ours. There are significant differences between our primary analysis and the re-analyzed data of Yang et al.; e.g., the different sample sizes and different data structures; these differences are described in the Supplementary methods in detail. Most notably, we had to repeat the analysis with the CanFam 3.1 dog reference genome, which was used by Yang et al. as well to obtain comparable results. Consequently, this analysis is appropriate for comparing our results to those published earlier, but they are slightly different from the results described above. After re-analyzing the data of Yang et al. [11], we found 12,435 expressed genes, somewhat more (+3.5%) than with our primary analysis. Figure 3 shows the gene count comparisons between the three analyses on a Venn diagram: (1) in red, the results published by Yang et al. (524 genes were published in their original paper, out of which 458 were retained in Ensembl’s 98th version used in this study); (2) in blue, the results we obtained by re-analyzing the data of Yang et al. [11] using our analysis pipeline and (3) in green, our analysis, which included an additional ten dog and eight wolf samples. Most of the differentially expressed genes published by Yang et al. [11] were also detected in the current study. However, we also identified an additional 1193 differentially expressed genes between dogs and wolves, which were not detected by Yang et al. [11]. Our re-analysis of the original dataset (two dogs vs. three wolves) of Yang et al. [11] could also identify most of the genes that were published in 2018.

### 3.4. Gene Ontology Analysis

A gene ontology (GO) analysis was implemented to investigate the functions of the differentially expressed genes. Figure 4 shows the most specific GO terms that were found significant (for the complete list of significant GO terms, see Appendix A). We used the set of all expressed genes in the blood (*n* = 12,014) as the background for this analysis.

In total, 136 gene ontology terms were found to be significant and those with the most extreme (either high or low) fold changes are shown in Figure 4. Gene ontology terms related to peptide and protein synthesis, the respiratory chain, and the centromeric region of chromosomes were enriched in the analysis.

We also performed gene ontology analyses of the top 500 most expressed genes in dogs and wolves separately. There were 366 overlapping genes in these two sets of genes. We identified some gene ontology terms enriched in dogs but not in wolves related to cell development (e.g., regulation of cell population proliferation, positive regulation of cell development or regulation of cell shape). Shared gene ontologies included, for example, actin-related ontologies (e.g., actin filament binding or regulation of actin filament polymerization) and immune system-related terms as well (e.g., in wolves: defense response; in dogs: immune response-activating cell surface receptor signaling pathway; regulation of T cell activation; in both: antigen processing and presentation; macrophage activation involved in immune response; Appendix A). Multiple gene ontologies related to gene expression were enriched in both species, for example, in dogs: regulation of gene expression, regulation of RNA biosynthetic process, RNA metabolic process; in wolves: regulation of DNA-templated transcription or rRNA binding; in both species: transcription regulator activity or structural constituent of ribosome.

### 3.5. Weighted Gene Correlation Network Analysis

To further investigate the biological differences between the wolf and dog gene expression profiles at the RNA level, we implemented a weighted gene correlation network analysis. As the 80% threshold in mean connectivity was not reached with the same parameter settings as used in the differential gene expression analysis for initial data filtration (minimum ten sequenced reads per gene in at least 11 sequenced animals), we increased the thresholds to meet the 80% requirement, as recommended by the WGCNA software’s authors. Finally, at minimum 10 sequenced reads in 20 or more animals were used for the WGCN analysis, resulting in 9799 expressed genes (a subset of the expressed genes, as defined for the differential gene expression analysis, containing 81.5% of the expressed genes), and accordingly, the soft-thresholding power was determined to be 24. The genes were divided into 18 gene co-expression modules, out of which 11 were significant at an alpha level of 5%, using the FDR-adjusted *p*-values estimated by the limma R package’s toptable() function (Figure 5A,B; Appendix A). These 11 modules contained clusters of co-expressed genes, some of which modules strongly correlated with each other as well as clustered together in a hierarchical clustering of their eigengenes ([25]; Figure 5C,D). We pooled together those genes that belonged to significant, highly positively correlated gene networks and obtained three joint modules from eight modules: M [1; 11], M [3; 5; 6] and M [2; 9; 12].

We then implemented a Reactome pathway analysis using these joint gene networks and identified significant pathways (Figure 5E). Most of the significant pathways were located in the metabolism of proteins, metabolism of RNA and cell cycle pathways. Many of these pathways were linked to gene transcription and translation, but some were related to, e.g., the immune system or to DNA replication.

## 4. Discussion

This study analyzed the poly-A tailed RNA fraction from the whole blood of 12 dogs and 11 wolves to identify the differentially expressed genes between dogs and their wild relatives, the grey wolves. We assumed that this analysis reveals how gene expression changed due to dog domestication.

We combined our European dataset with the Asian dataset of Yang et al. [11] and Liu et al. [13]. The dog samples of this current study were collected in two geographically distinct locations and represented one breed of European origin (*n* = 10 from Hungary) and two individuals with unknown ancestry from China, Asia. The sequencing depth of the European samples was considerably higher than that of the Asian samples. However, it was sufficiently high even in the Asian samples for a standard differential gene expression analysis, according to the ENCODE standards [28].

As expected, a large proportion of the reads aligned to hemoglobin genes. The large variance of the hemoglobin reads between the samples (1–55%, Table 2) was, however, unexpected. Part of this variation might be due to the possible differences in blood sampling and processing between the different studies. However, since the same high variation can be observed within groups of either studies or samples of geographical locations, this is unlikely to explain most of the variation in the hemoglobin-related read counts. This high read count variation led to an uneven decrease in the number of reads after the in silico removal of the hemoglobin reads, as Harrington et al. [10] advised.

Out of the 20,567 known canine protein-coding-genes, 12,014 (58%) were expressed in at least 11 out of the 23 analyzed samples. Following the analysis of the rlog-transformed read counts, we could conclude that both the geographical origin (termed in the manuscript as population effect) and the species had a significant effect on the gene expression profile of the samples. The population effect is indicated by the multidimensional scaling analysis (Figure 1), the principal component analysis (Appendix A) and the estimated Euclidean and Poisson distances between the animals (Appendix A). When a third population of wolves from the American continent was also included in the analysis (Appendix A) the population effect persisted, reinforcing our hypothesis that gene expression patterns in whole blood change as a function of geographical location. This is in accordance with the recent findings of Hoffman et al. [29]. Recently Leathlobhair et al. [30] showed based on mitochondrial sequence data of dogs spanning ~9000 years that North American dogs have Eurasian ancestry. This can explain why we see the North American wolves as a distinct outlier group in Appendix A and since it was shown in the same study that native American dogs were almost completely extinct, we hypothesize that if Appendix A were extended with more samples in the future—including North American dogs as well—we would still see the North American wolves as a separate, distinct cluster, farther away from the other dog breeds and wolves than any other cluster. Finally, on the MDS plot the two analyzed Asian dogs clustered closer to the Asian wolves than to the European dog breed, although, this clustering disappeared on the MDS plot with the American wolves included (Appendix A).

Whole blood has heterogeneous cell populations and is regularly used for diagnostic purposes in the veterinary field. Its chemical parameters should stay in a narrow range, but its harmonic composition is disturbed by the environment regularly [19]. Gene expression is a slower process and more resistant to slight impacts, but in our case, the distance in space and time is so massive between the above-mentioned populations, that it can easily cause the observed population effect. Regarding the present analysis, the population effect (i.e., the geographical origin of the samples) was included in the model, allowing us to avoid the detection of false positive DEGs as a consequence of this effect in the dog–wolf comparison.

We note that the population effect cannot be clearly differentiated from an unknown breed effect. We also considered the possibility to include a breed effect in the model. Unfortunately, several issues could not be addressed given our sample composition: firstly, no breed information is available on the Chinese indigenous dogs, and therefore, grouping them into the same breed group might be an equally large mistake as separating them into different breed groups (*n* = 2 sample size would be also insufficient to estimate statistical parameters too). The same applies to the 11 wolves: some were from Tibet (southern China) and some from Inner Mongolia (northern China). Their clustering into a pseudo-wolf breed might be a mistake as well since significant differences might be present between the samples and the underlying wolf populations. Consequently, including a breed effect in the analysis was not feasible at the present time. However, this question is of scientific interest and should be addressed in the future when our analysis is further extended by the broader scientific community.

In the presented MDS, PCA and distance analyses, it is visible that both the dog and wolf populations are diverse. The diverse populations are an advantage of this study, because the higher within-group variance (which is also prominent in the dog cluster once the European dogs are included) helps detect those genes that are indeed differentially expressed between the two species.

### 4.1. Comparisons with the Results of Yang et al. [11]

Due to the overlapping data and analysis goals between our study and that of Yang et al. [11], it was natural to compare the results of the two studies. Several major differences existed between the two studies, however. Most notably, the annotation of the canine reference genome had changed, which introduced some differences on the reference genome’s side. Another issue is the partly different analysis pipeline. For example, we used the HISAT2 v.2.0.0 aligner software, which is the successor of the tophat aligner used by Yang et al. [11]. Software version differences (e.g., in the DESeq2 v.1.44.0 R package) also existed, all of which contributed to the fact that the exact same results could not be reproduced as in 2018 [11]. Out of the 524 differentially expressed genes identified by Yang et al. [11], we found 458 in the Ensembl v98 annotation file, which we used for the CanFam 3.1 reference genome version. Therefore, this gene set formed the basis of comparisons between the two studies.

Yang et al. [11] identified 18925 expressed genes, which is considerably higher than the 14,038 expressed genes identified here. This is primarily due to the different expressed gene definitions in the two studies: Yang et al. called an expressed gene at the minimum of one read per gene, while this threshold was ten reads in at least three individuals in our study (the required minimum number of individuals to carry the gene at the specified level was not reported by Yang et al.). Another possible explanation is the fact that we removed the hemoglobin-related genes as recommended by Harrington et al. [10], which in silico filtering was not applied in Yang et al. [11].

We also re-analyzed the dataset of Yang et al. [11], excluding the European dogs. This led to an analysis that is more similar to that published by Yang et al. [11], facilitating the comparison between the two studies. Figure 3 compares the differentially expressed genes between the three analyses. We identified most of the DEGs identified by Yang et al. [11]. However, we found significantly more differentially expressed genes.

The outlined methodological differences could have also contributed to the significantly higher number of identified DEGs here (1576 vs. 458). We found 377 DEGs identified by Yang et al. [11] (or approximately 82%), which is a high proportion, given the cardinal differences between the methods of the two studies. When the same dataset was analyzed, as in Yang et al. [11], an additional 44 DEGs out of the 458 published in 2018 were identified, making the overlap between the two studies ~92% in terms of DEGs. By re-analyzing the original data of Yang et al., we identified 411 new DEGs compared to the published results of Yang et al., which is a result of the different α value used here (0.1 instead of the 0.05 reported by Yang et al. [11]). However, according to the scripts shared with us by Yang et al. [11], they applied the 5% threshold in the *subset()* function of DESeq2 and used the default 10% with the *results()* function, which is against the recommendations of the authors [22].

### 4.2. Functional Gene Ontology Analysis of the Differentially Expressed Genes

Many of the enriched gene ontologies are related to protein synthesis and gene translation (for example, cytosolic large ribosomal subunit, cytoplasmic translation, or DNA-binding transcription factor activity, RNA polymerase II-specific). However, based on these GO terms, no specifics are known about those proteins, making it challenging to conclude their link to domestication. Sahlén et al. [31], who investigated the role of cis-regulatory elements in dog domestication, identified promoter and enhancer regions, which included SNPs with highly differentiated allele frequencies between wolves and dogs. They concluded that positive selection for specific SNP alleles located in cis-regulatory elements led to highly differentiated gene expression levels between dogs and wolves which was most likely crucial for domestication. This also suggests that alterations in the regulation of gene expression might be a fundamental genetic aspect of a domestication process. Therefore, the altered expression of several genes linked to translational processes detected in our dataset is in accordance with this previous finding, supporting the hypothesis about the pivotal role of gene expression regulation in domestication. Although the two analyses are very different methodologically, they indicate that both pre- and post-transcriptional regulation had an essential role in domestication.

To better understand the functional role of genes expressed in the blood of dogs and wolves, we implemented a second gene ontology analysis. We examined the gene ontologies of the top 500 most expressed genes (apart from the hemoglobin genes, which were excluded earlier) in both species separately by using the Slim database of the PantherDB on-line tool. We found that most of these genes (*n* = 366 or 73%) overlapped between the two species. This also explains why 26 of the enriched gene ontology terms were shared between them. We found 42 and 62 enriched GO terms in wolves and dogs, respectively. The significant GO terms in this analysis also support that gene expression regulation might play have had an important role in canine speciation (e.g., transcription regulator activity) with the other gene ontology analysis, but gene ontologies related to the immune system (e.g., macrophage activation involved in immune response or antigen processing and presentation).

### 4.3. Gene Correlation Networks

A more stringent gene filtering had to be applied to the WGCN analysis in order to comply with the recommendations. This resulted in a total of 9799 expressed genes, which is ~81.5% of the genes called during the differential gene expression analysis. The per-individual threshold was the same for the two analyses (an expressed gene with a minimum of 10 reads was called in any individual), indicating that even in the differential gene expression analysis, most of the genes were expressed in at least 20 individuals out of 23.

In total, 18 gene modules were differentiated out of which 11 were significantly different between the two species (Figure 5). These correlation networks hosted 4402 genes in total, approximately 45% of all tested genes. Many significant pathways are related to gene expression and/or gene expression regulation either directly or indirectly, supporting our hypothesis, that gene expression regulation played a crucial role in dog domestication. Some pathways were overrepresented in one module, but underrepresented in the other. For example, the translation initiation complex formation pathway was overrepresented in the [3; 5; 6] gene network (fold enrichment: 5.44), while it was underrepresented in the [1; 11] gene network (fold enrichment: <0.01). This is a result of the very negative correlation between these networks (Figure 5C). The L13a-mediated translational silencing of Ceruloplasmin expression pathway is another example, with a role in translational splicing. We also identified many significant pathways related to cell cycle, metabolism or DNA replication for example, but their role in speciation is yet unknown.

Although we cannot readily prepare causative relationships between these biological pathways and the domestication process of the dog based on gene ontology and genetic pathway analyses of RNA sequencing data, these results suggest that a wide range of molecular characteristics of the dogs have changed compared to their wild ancestors. Some of these changes were most likely crucial for domestication itself, although it is yet unknown which ones.

One limitation of this study is that we could include dogs from only one additional dog breed, from border collie in the analysis in addition to the Chinese feral dogs and therefore the dog population is still not as diverse as it could be. Regarding the wolf samples, we relied on publicly available sequence data and therefore we had no influence on sample collection. We believe that in the future the further extension of this research will help us shed more light on dog domestication and help us disentangle the roles of the outlined genetic pathways. For this, we recommend that researchers focus on the further diversification of the dog and wolf populations in the future from both genetic and geographical aspects without merely focusing on increasing the sample sizes.

## 5. Conclusions

We identified 1576 differentially expressed genes between dogs and wolves and 4402 genes in 11 gene networks. Based on the gene ontology analysis of the significantly differentially expressed genes and the pathway analysis of the significant gene networks, we hypothesize that gene expression regulation might have played a role in the domestication of the dog. This hypothesis can be further investigated in the future by analyzing other domesticated species and their close, wild relatives.

## Figures and Tables

**Figure 1 animals-14-02655-f001:**
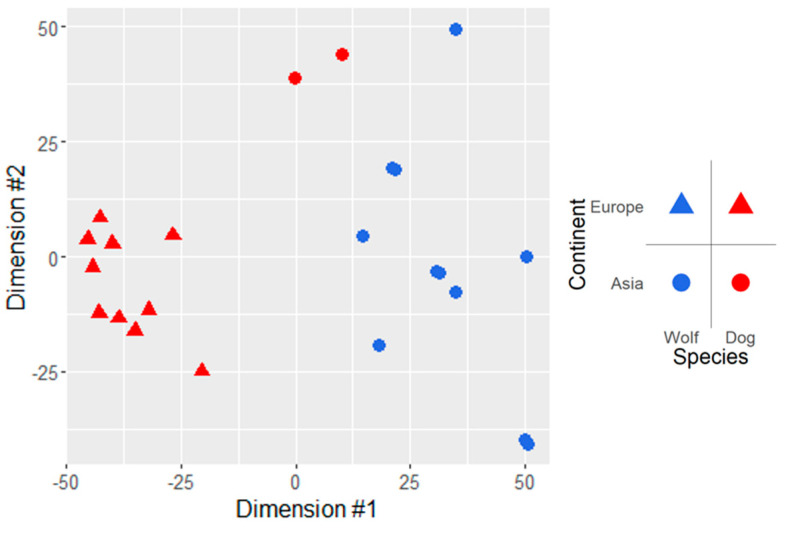
Multidimensional scaling plot of the regularized−logarithm transformed raw read counts. Shapes indicate the origin of the samples.

**Figure 2 animals-14-02655-f002:**
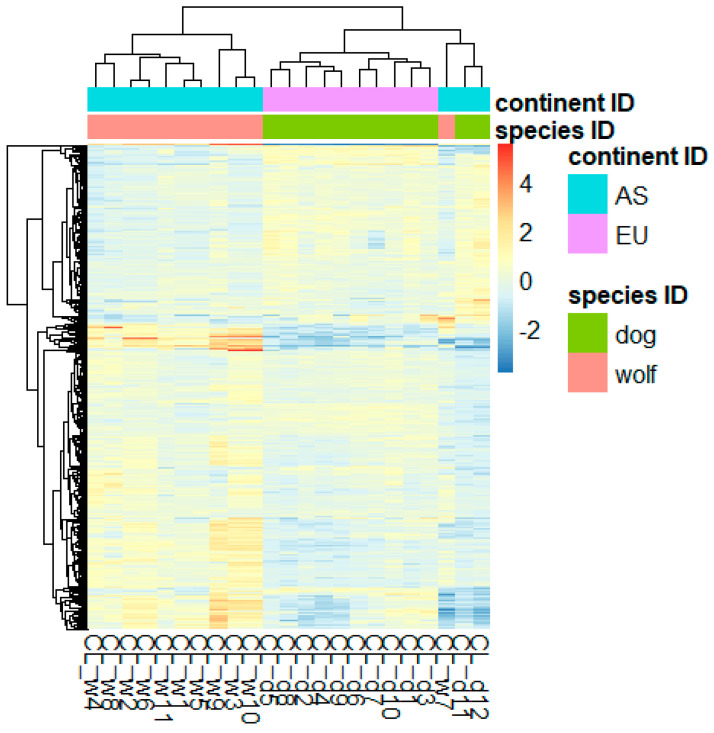
Heatmap of the differentially expressed genes (DEG). Clustering of subjects (x axis; top) and genes (y axis; left) based on gene expression levels of the DEGs are shown at the margins of the heatmap.

**Figure 3 animals-14-02655-f003:**
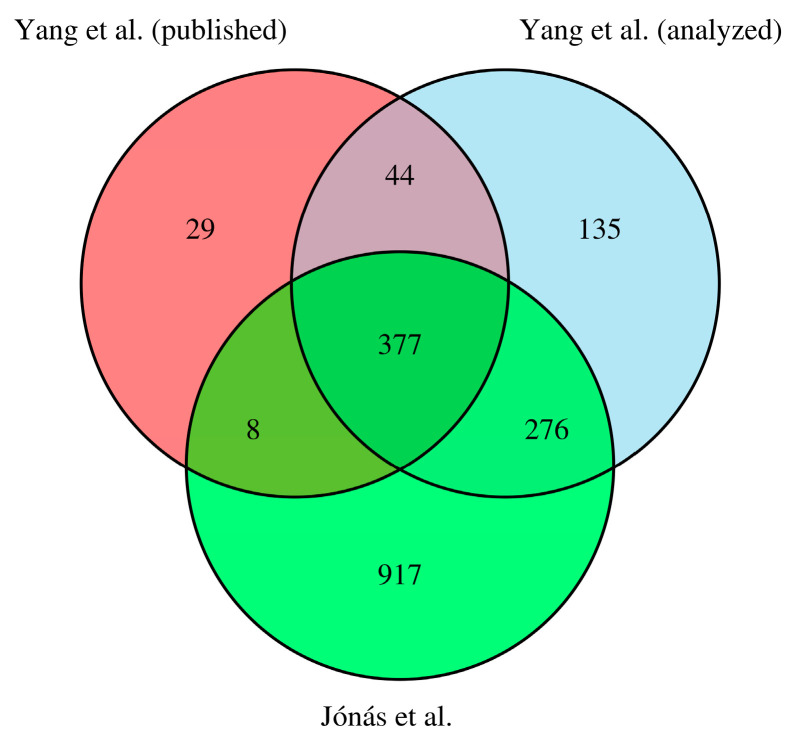
A Venn diagram of the number of differentially expressed genes detected in this study (green) with those published by Yang et al. [11] (red) and the re-analyzed data of Yang et al. [11] (blue).

**Figure 4 animals-14-02655-f004:**
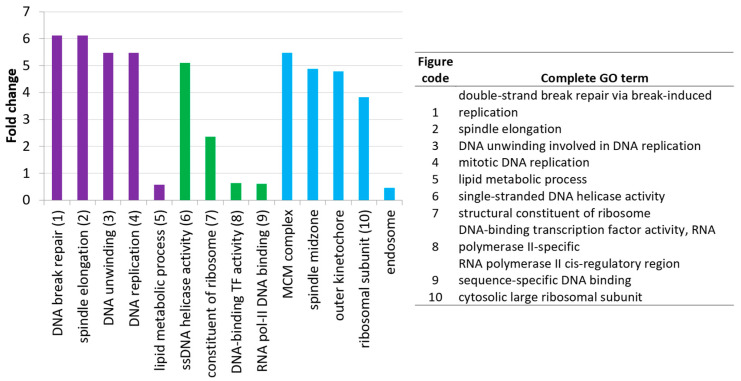
Gene ontology analysis of 1576 significantly differentially expressed genes. Background gene set: all expressed genes. Coloring corresponds to the three primary gene ontology categories: purple—biological process; green—molecular function; blue—cellular components. Only the GO terms with the highest and lowest fold changes are shown for each category.

**Figure 5 animals-14-02655-f005:**
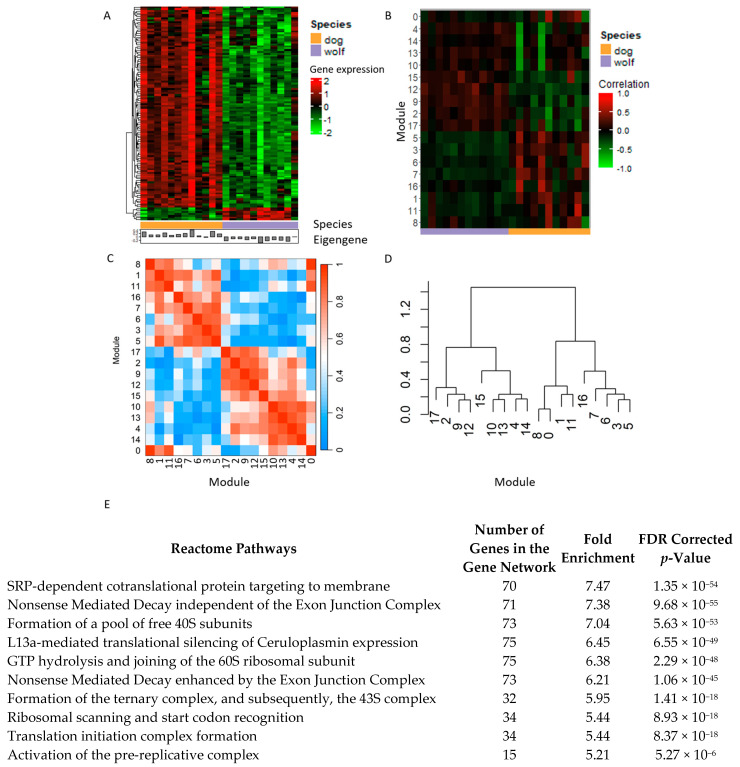
Summary results of the weighted gene correlation network analysis. (**A**) Gene expression heatmap of the most significant gene network (module #12); dogs and wolves are on the x axis, and genes of the network (*n* = 115) are on the y axis. Heatmap colors represent gene expression levels in the 23 tested individuals. (**B**) Heatmap showing the relationship between treatment groups (wolf-dog; x-axis) and gene network modules (y-axis). (**C**) Eigengene heatmap showing the correlation between the 18 modules and (**D**) the dendrogram of hierarchical clustering of the eigengenes of each module. (**E**) Ten significant reactome pathways. Selection criteria: minimum of ten genes of the given pathway had to be presented in the joint gene network and with the highest fold enrichments as compared to the background of all 9799 genes.

**Table 1 animals-14-02655-t001:** Additional information about the 23 analyzed samples.

ID	Species/Breed ^1^	Sample Origin (Country) ^2^	SRA Run ID	Source
CL_d1	Dog/BC	EU (Hungary)	SRR18645125	[12]
CL_d2	Dog/BC	EU (Hungary)	SRR18645124	[12]
CL_d3	Dog/BC	EU (Hungary)	SRR18645121	[12]
CL_d4	Dog/BC	EU (Hungary)	SRR18645120	[12]
CL_d5	Dog/BC	EU (Hungary)	SRR18645122	[12]
CL_d6	Dog/BC	EU (Hungary)	SRR18645119	[12]
CL_d7	Dog/BC	EU (Hungary)	SRR18645123	[12]
CL_d8	Dog/BC	EU (Hungary)	SRR18645118	[12]
CL_d9	Dog/BC	EU (Hungary)	SRR18645117	[12]
CL_d10	Dog/BC	EU (Hungary)	SRR18645116	[12]
CL_d11	Dog/ICD	AS (China, Henan pr.)	SRR5025823	[11]
CL_d12	Dog/ICD	AS (China, Henan pr.)	SRR5025782	[11]
CL_w1	Wolf/-	AS (China, Tibet)	SRR5026184	[11]
CL_w2	Wolf/-	AS (China, Tibet)	SRR5026185	[11]
CL_w3	Wolf/-	AS (China, Inner Mongolia)	SRR5026249	[11]
CL_w4	Wolf/-	AS (China)	SRP093411	[13]
CL_w5	Wolf/-	AS (China)	SRP093423	[13]
CL_w6	Wolf/-	AS (China)	SRP093543	[13]
CL_w7	Wolf/-	AS (China)	SRP093547	[13]
CL_w8	Wolf/-	AS (China)	SRP093548	[13]
CL_w9	Wolf/-	AS (China)	SRP093549	[13]
CL_w10	Wolf/-	AS (China)	SRP093550	[13]
CL_w11	Wolf/-	AS (China)	SRP093551	[13]

^1^: BC—border collie, ICD—indigenous Chinese dog; ^2^: EU—Europe, AS—Asia, pr.—province. Charruau et al. (2016) [14].

**Table 2 animals-14-02655-t002:** Basic alignment statistics of the 23 analyzed samples.

Sample ID	Number of Sequenced Reads	Number of Reads after Adapter Trimming	Number of Aligned Reads		Number of Hemoglobin Reads	
			N	%	N	%
CL_d1	139,574,092	99,781,996	88,729,873	88.92	46,563,726	52.48
CL_d2	145,101,716	114,231,400	101,985,098	89.28	16,884,738	16.56
CL_d3	211,298,362	137,241,346	122,636,400	89.36	67,116,393	54.73
CL_d4	115,495,360	102,506,414	89,562,963	87.37	14,331,833	16.00
CL_d5	100,769,724	76,546,858	67,690,141	88.43	27,145,508	40.10
CL_d6	132,309,048	100,814,174	89,499,489	88.78	25,338,246	28.31
CL_d7	115,616,142	93,060,894	81,667,714	87.76	21,381,691	26.18
CL_d8	120,024,014	91,619,710	80,658,649	88.04	23,235,076	28.81
CL_d9	124,942,900	103,191,914	90,555,704	87.75	21,889,720	24.17
CL_d10	148,054,394	103,245,778	93,220,105	90.29	37,326,335	40.04
CL_d11	42,671,378	42,671,378	40,868,038	95.77	4,477,100	10.96
CL_d12	49,896,210	49,896,210	47,664,640	95.53	1,382,724	2.90
CL_w1	50,597,914	50,597,914	48,077,660	95.02	12,436,555	25.87
CL_w2	50,190,664	50,190,664	47,853,645	95.34	14,554,543	30.41
CL_w3	47,960,542	47,960,542	46,232,389	96.40	11,531,137	24.94
CL_w4	44,621,354	44,621,354	42,183,724	94.54	10,245,046	24.29
CL_w5	50,597,914	50,597,914	47,774,933	94.42	12,267,628	25.68
CL_w6	50,190,664	50,190,664	47,549,364	94.74	14,367,872	30.22
CL_w7	42,506,574	42,506,574	39,606,586	93.18	287,597	0.73
CL_w8	43,749,010	43,749,010	41,982,200	95.96	8,143,261	19.40
CL_w9	45,656,890	45,656,890	43,311,096	94.86	14,273,187	32.96
CL_w10	47,960,542	47,960,542	45,959,640	95.83	11,389,578	24.78
CL_w11	47,755,968	47,755,968	45,226,589	94.70	10,221,679	22.60
Average dog	120,479,445	92,900,673	82,894,901	0.90	25,589,424	28.44
Average wolf	47,435,276	47,435,276	45,068,893	0.95	10,883,462	23.81

## Data Availability

Raw RNA sequence data are available at the Sequence Read Archive (SRA) of the [American] National Center for Biotechnology Information (NCBI) under SRA ID: SRP367668 and in the case of the data of Yang et al. under the following IDs: SRP091691, SRP093404, SRP093423, SRP093543 and SRP093550. All intermediate data generated during the analysis (e.g., read count matrix, differentially expressed gene list, “data behind the figures”, etc.) are deposited at *Science Data Bank* (https://www.scidb.cn) with the following DOI accession number: 10.57760/sciencedb.09565. The data of American wolves used to generate Appendix A and published earlier by Charruau et al. [14] are available at NCBI’s Gene Expression Omnibus, under the GSE80440 ID. Code Availability: All log files and scripts used for data analysis are publicly available on *Science Data Bank* (https://www.scidb.cn) with the following DOI accession number: 10.57760/sciencedb.09565. In case of questions, please contact the corresponding author. Appendix A and other intermediate files produced during the data analysis are available on *Science Data Bank*, under the same DOI accession number.

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
