# Peer review of "Dog Domestication Strongly Relied on Translation Regulation According to Differential Gene Expression Analysis"

_animals, 2024, doi:10.3390/ani14182655_

Round 1

Reviewer 1 Report

Comments and Suggestions for Authors

The topic of the paper is interesting and I think has to the potential to be an impactful article with revision, including:

1) Rewriting the introduction with more attention to the broader aim of the paper, which from the title seems to be to investigate the transcriptomic differences between dogs and wolves and potentially the transcriptomic signature of domestication. Instead, the introduction seems to overly focus on describing general methodologies (such as RNA-seq) and describing a single paper in considerable detail (i.e., Yang et al. 2018). Finally, I think the authors should consider rethinking the aim of this paper as described in the introduction. From the current aim, it seems like the paper is a slightly expanded version of Yang et al. study. I would recommend the authors develop an aim that is more refined and focused. For example, what questions or hypotheses can you generate with your data that are different from those in Yang et al.?

2) The methods need to include a more specific description of how all blood samples were collected, preserved, prepped (e.g., library prep), and sequenced. All of these steps can impact the downstream data and should be transparent to the reader. One way to handle describing these steps would be through a table that provides a side-by-side comparison. Moreover, I would consider expanding this table to include the main steps performed through the differential gene expression analysis, so that readers can compare similarities and differences between each dataset and analyses.

3) The authors should consider expanding the functional analyses. Possibly more rigorous pathway analyses, gene correlation network analyses, etc. I think this data has the potential to be more informative than just a list of significant genes and gene ontology terms.

4) Similarly, I think the authors should consider expanding their dataset. This paper is already performing a meta-analysis by combining samples from multiple experiments. There are other public RNA-seq experiments on dogs that could potentially be included to make this analysis more robust.

5) Similar to the introduction, the authors should consider revisiting the discussion and conclusion. I appreciate the authors commitment to explaining some of the expected and possibly unexpected issues or observations in their data and analysis (e.g., the variation in hemoglobin reads among samples or the detailed differences between their analysis and results and that of Yang et al. 2018), but I felt there was comparatively little synthesis and interpretation of your functional analyses. Moreover, it was unclear to me how simply identifying differentially expressed genes is addressing the hypothesis put forth in the conclusion that 'gene expression regulation might have played a role in the domestication of the dog' (line 424). 

Comments on the Quality of English Language

Quality of writing was reasonable. No major grammatical issues were observed.

Author Response

Reply to Reviewer I.

We are grateful for the reviewer’s comments and valuable recommendations. In the following, we show a detailed list of responses written in blue.

Comments and Suggestions for Authors

The topic of the paper is interesting and I think has to the potential to be an impactful article with revision, including:

Reply: We would like to thank the Reviewer for recognizing the importance of our work.

1) Rewriting the introduction with more attention to the broader aim of the paper, which from the title seems to be to investigate the transcriptomic differences between dogs and wolves and potentially the transcriptomic signature of domestication. Instead, the introduction seems to overly focus on describing general methodologies (such as RNA-seq) and describing a single paper in considerable detail (i.e., Yang et al. 2018). Finally, I think the authors should consider rethinking the aim of this paper as described in the introduction. From the current aim, it seems like the paper is a slightly expanded version of Yang et al. study. I would recommend the authors develop an aim that is more refined and focused. For example, what questions or hypotheses can you generate with your data that are different from those in Yang et al.?

Reply: Corrected. The introduction is completely re-written and the aims are re-phrased. Now we emphasize that our aim was to do an original research (including methods not used on these data before us), following up the analysis of Yang et al.

2) The methods need to include a more specific description of how all blood samples were collected, preserved, prepped (e.g., library prep), and sequenced. All of these steps can impact the downstream data and should be transparent to the reader. One way to handle describing these steps would be through a table that provides a side-by-side comparison. Moreover, I would consider expanding this table to include the main steps performed through the differential gene expression analysis, so that readers can compare similarities and differences between each dataset and analyses.

Reply: Done. Because of its size, the table is put in the supplementary materials.

3) The authors should consider expanding the functional analyses. Possibly more rigorous pathway analyses, gene correlation network analyses, etc. I think this data has the potential to be more informative than just a list of significant genes and gene ontology terms.

Reply: Done. We found this recommendation exceptionally useful. After implementing a weighted gene correlation network analysis, we identified 11 significant gene co-expression networks and examined those networks in a genetic pathway analysis. These results provided further support to our differential gene expression analysis results.

4) Similarly, I think the authors should consider expanding their dataset. This paper is already performing a meta-analysis by combining samples from multiple experiments. There are other public RNA-seq experiments on dogs that could potentially be included to make this analysis more robust.

Reply: Done. We checked the scientific literature with available sequence data and found eight more wolves to be included (Liu et al., 2017). We also considered other published data, but found only data that implemented hemoglobin extraction before sequencing (we considered again the wolves from Charruau et al., 2016 and the control dogs from Borchert et al., 2020). We didn’t want to include samples that were so differently treated in the experimental laboratory compared to the other analyzed data.

Our results didn’t change much after the inclusion of eight more wolves and the significant genes were 75% the same as before, the rest of the genes had low fold changes in the initial analysis (i.e. in the original submission, including only three wolves), which were therefore likely to be false positives. ~300 new genes were found to be significant after including more wolf samples.

At this point, we would like to point out that we had to re-do the entire analysis from the alignment to the reference genome step, because the official online tool for gene ontology analysis (pantherDB; link: https://pantherdb.org) was updated and does not support the earlier CanFam 3.1 canine genome version any longer. Therefore, we switched to the more recent ROS Cfam 1.0 dog reference genome and to the pantherDB’s most recent version (v19); all this information was updated in the manuscript.

5) Similar to the introduction, the authors should consider revisiting the discussion and conclusion. I appreciate the authors commitment to explaining some of the expected and possibly unexpected issues or observations in their data and analysis (e.g., the variation in hemoglobin reads among samples or the detailed differences between their analysis and results and that of Yang et al. 2018), but I felt there was comparatively little synthesis and interpretation of your functional analyses. Moreover, it was unclear to me how simply identifying differentially expressed genes is addressing the hypothesis put forth in the conclusion that 'gene expression regulation might have played a role in the domestication of the dog' (line 424).

Reply: Done. The discussion section was updated; we included discussion on the new results, revised the previous discussion section and rephrased it wherever we deemed it necessary. We hope it is better. The conclusion section was almost completely re-written partly also to comply with the second Reviewer’s request to make it shorter.

Reviewer 2 Report

Comments and Suggestions for Authors

The Authors analyzed a set of 10 Border collie samples. They re-analyzed the samples of a previous paper of Yang et al. 2018. This resulted in a combined dataset of 12 dogs (two dogs from Yang et al.2018 and ten dogs) and three wolf samples from Yang et al. 2018.

Why the Authors chose 10 Border collie? Please explain. I do not understand if the 10 samples are from the present study or from the study of Jónás et al. submitted cited in the manuscript as reference.

The supplementary materials are missed and is impossible for me understand some results. Furthermore, I do not understand if the supplementary data in the repository Science Data Bank are from this paper or by Jonas, et al (submitted). What are the differences between the two papers? Please explain.

Line 12: please change “Previously, Yang et al. (2018, Anim. Genet. 49:291–302).....” with “Previously, Yang et al. [15].....”

Line 62, 99, 190. 225 and so on: pay attention to the reference citation format.

Line 88: please change “Jónás et al. (submitted) [16]….” with “Jónás et al. (present study and submitted) [16]….”

Figure 1: the colors do not match the legend: “continent ID” where are black AS and EU in the figure?

247: Supplementary methods are not provided. Please add the mentioned differences about your data and Yang et al.. Otherwise, the sentence have to be removed.

Figure 3 legend: please change “A 3-fold comparison…..” with “A Venn diagram…..”

The “conclusion section” in my opinion is too long and have to be reduced.

Author Response

Reply to Reviewer II.

We thank the reviewer’s comments. Our answers to each recommendation below are written in blue.

Comments and Suggestions for Authors

The Authors analyzed a set of 10 Border collie samples. They re-analyzed the samples of a previous paper of Yang et al. 2018. This resulted in a combined dataset of 12 dogs (two dogs from Yang et al.2018 and ten dogs) and three wolf samples from Yang et al. 2018.

Why the Authors chose 10 Border collie? Please explain. I do not understand if the 10 samples are from the present study or from the study of Jónás et al. submitted cited in the manuscript as reference.

Reply: The samples are shared between the two Jónás et al. studies, i.e. they were used for both and the two manuscripts are planned to be published at around the same time. The aim of the elsewhere submitted Jónás et al. article was to investigate the effect of age on the gene expression levels and we wanted to use only one breed to avoid the possible breed effect as a confounding factor (if any difference was to be found, the study could have been continued by the inclusion of other species later, in an attempt to generalize our findings). The border collie species was a practical choice, as it is very popular among dog owners and available in large enough numbers (both young and older animals).

The supplementary materials are missed and is impossible for me understand some results. Furthermore, I do not understand if the supplementary data in the repository Science Data Bank are from this paper or by Jonas, et al (submitted). What are the differences between the two papers? Please explain.

Reply: The referenced Supplementary data 1, 2 and “supplementary_materials_20240715.docx” files were uploaded as a single, compressed folder, as requested by the journal. We don’t know why the Reviewer couldn’t access these supplementary materials, but we hope that in the second round of the review, they will not have a similar problem. The files uploaded to the Science Data Bank repository are for the present article.

Differences between the two papers: in the other, submitted manuscript, the effect of age on RNA expression levels in whole blood, only in border collie dogs (i.e. without wolves) is examined.

Line 12: please change “Previously, Yang et al. (2018, Anim. Genet. 49:291–302).....” with “Previously, Yang et al. [15].....”

Reply: Corrected.

Line 62, 99, 190. 225 and so on: pay attention to the reference citation format.

Reply: Corrected.

Line 88: please change “Jónás et al. (submitted) [16]….” with “Jónás et al. (present study and submitted) [16]….”

Reply: Corrected.

Figure 1: the colors do not match the legend: “continent ID” where are black AS and EU in the figure?

Reply: Corrected.

247: Supplementary methods are not provided. Please add the mentioned differences about your data and Yang et al.. Otherwise, the sentence have to be removed.

Reply: The “supplementary_materials_20240715.docx” file included the short supplementary methods section, which was uploaded, as we wrote above, in a compressed folder.

Figure 3 legend: please change “A 3-fold comparison…..” with “A Venn diagram…..”

Reply: Corrected.

The “conclusion section” in my opinion is too long and have to be reduced.

Reply: Corrected. In addition, the introduction, discussion and conclusion sections were significantly re-written to focus more on the scientific topic. During this re-write process, we wrote a shorter conclusion section.

Reviewer 3 Report

Comments and Suggestions for Authors

The authors endeavor to integrate and analyze their own and other scholars' published data in order to generate novel insights. While the utilization of open source data is commendable, there are certain challenges that need to be addressed in this study.

1. As depicted in Table 1, the researchers of the dataset referenced in [15] sought to identify disparities in gene expression between Asian wolves and indigenous Chinese dogs. The authors of the present study augmented the original dataset with a larger sample size for comparison, potentially introducing new biases into the results.

2. As illustrated in Table 2, the number of sequences sequenced from datasets cited in references [15], [16], and [23] as well as the resulting comparison rates with genomes do not exhibit sufficient alignment, which could impact the reliability of comparative outcomes.

3. The PCA analysis findings presented in Figure 1 indicate clear differentiation among European dogs, Asian dogs, and wolves; however, it is conceivable that issues mentioned in points 1 and 2 above may have contributed to insufficient result reliability. The identified differential genes between wolves and dogs lack adequate credibility, necessitating further validation through additional testing.

4. Other analytical findings raise concerns akin to those observed in Figure 1.

In conclusion, it is recommended that the authors broaden their data collection scope and enhance the quality of their analytical dataset to ensure scientific rigor within their results. Additionally, supplementing these findings with verification results from actual animals would be beneficial. Given that collecting blood samples from wolves and dogs is relatively feasible, substantial result deviations should not be tolerated lightly.

Comments on the Quality of English Language

Minor editing of English language required

Author Response

Reply to Reviewer III.

We thank the reviewer’s work and recommendations. Below we try to address every issue (our replies will be in blue color).

Comments and Suggestions for Authors

The authors endeavor to integrate and analyze their own and other scholars' published data in order to generate novel insights. While the utilization of open source data is commendable, there are certain challenges that need to be addressed in this study.

  1. As depicted in Table 1, the researchers of the dataset referenced in [15] sought to identify disparities in gene expression between Asian wolves and indigenous Chinese dogs. The authors of the present study augmented the original dataset with a larger sample size for comparison, potentially introducing new biases into the results.

Reply: We agree that including only border collies from Hungary introduced a new bias and we acknowledged this limitation in the manuscript

  1. As illustrated in Table 2, the number of sequences sequenced from datasets cited in references [15], [16], and [23] as well as the resulting comparison rates with genomes do not exhibit sufficient alignment, which could impact the reliability of comparative outcomes.

Reply: Alignment rate to the ROS Cfam 1.0 reference genome is 90% in dogs and 95% in wolves (slightly higher than to the initially used CanFam 3.1 reference genome). In the revised version, we updated the entire analysis to the current reference genome, and we believe this alignment rate is sufficiently high. ENCODE recommendations for polyA-capture RNA sequencing is 30 million sequenced fragments (https://www.encodeproject.org/about/experiment-guidelines/; please see the “ENCODE Experimental Guidelines for ENCODE3 RNA-seq”).

  1. The PCA analysis findings presented in Figure 1 indicate clear differentiation among European dogs, Asian dogs, and wolves; however, it is conceivable that issues mentioned in points 1 and 2 above may have contributed to insufficient result reliability. The identified differential genes between wolves and dogs lack adequate credibility, necessitating further validation through additional testing.

Reply: Figure 1 presented multidimensional scaling (MDS) analysis results and not PCA. PCA was presented in the supplementary data (Figure S2). We addressed many previous limitations in the manuscript and we hope that this point is also properly dealt with.

We also would like to emphasize what the Reviewer also mentioned above: “The authors endeavor to integrate and analyze their own and other scholars' published data in order to generate novel insights”. A great challenge in bioinformatics is that the enormous amount of data (e.g. currently 12 petabytes of sequence data is available only on NCBI’s Sequence Read Archive database) generated by biologists needs to be analyzed and a lot of knowledge still “lies within the data”, waiting to be explored. The “recycling” or reuse of the already published data is answering this challenge. However, an important inherent disadvantage of such bioinformatic studies is that the scientists do not have influence on the data generation process. This is clearly a limitation of this study too, which we now further emphasized in our manuscript. However, we believe our results are still advancing our knowledge in this field of science and is a significant contribution to the understanding on the domestication history of the dog.

  1. Other analytical findings raise concerns akin to those observed in Figure 1.

Reply: We improved considerably all analyses and the presented results were updated additionally, following the more detailed guidelines of the other Reviewers. We hope we could answer all concerns of the Reviewer; in case not, please elaborate the problematic issues and we will answer them.

In conclusion, it is recommended that the authors broaden their data collection scope and enhance the quality of their analytical dataset to ensure scientific rigor within their results. Additionally, supplementing these findings with verification results from actual animals would be beneficial. Given that collecting blood samples from wolves and dogs is relatively feasible, substantial result deviations should not be tolerated lightly.

Reply: We included eight more wolf samples in our dataset (from Liu et al., 2017) and tried to include even more wolves and dogs (e.g. from Charruau et al., 2016 and the control dogs from Borchert et al., 2020), however, we were unable to find more appropriate data, which would match the analyzed whole blood transcriptomes. If the Reviewer would know any dataset that has the same parameterization – please see the new Table S2 for details – we would be grateful for the recommendations.

Round 2

Reviewer 2 Report

Comments and Suggestions for Authors

The authors addressed most of the reviewer concern and made appropriate changes to the manuscript. I think the questions raised were now satisfactorily answered.

Reviewer 3 Report

Comments and Suggestions for Authors

 I noticed that the authors' efforts greatly increased the overall credibility of the paper's conclusions. This increase can be due to the more comprehensive review of the relevant literature, the stricter methodological approach used in data collection and analysis, and the clear display of the findings. Moreover, the authors skillfully deal with possible objections and limitations in the study, thus enhancing the validity of their presentation.

Comments on the Quality of English Language

Minor editing of English language required.